# GRAPH PERMUTATION SELECTION FOR DECODING OF ERROR CORRECTION CODES USING SELF-ATTENTION

## ABSTRACT

Error correction codes are an integral part of communication applications and boost the reliability of transmission. The optimal decoding of transmitted code-words is the maximum likelihood rule, which is NP-hard. For practical realizations, suboptimal decoding algorithms are employed; however, the lack of theoretical insights currently impedes the exploitation of the full potential of these algorithms. One key insight is the choice of permutation in *permutation decoding*. We present a data-driven framework for permutation selection combining domain knowledge with machine learning concepts such as node embedding and self-attention. Significant and consistent improvements in the bit error rate are shown for the simulated Bose Chaudhuri Hocquenghem (BCH) code as compared to the baseline decoders. To the best of our knowledge, this work is the first to leverage the benefits of self-attention networks in physical layer communication systems.

## 1 INTRODUCTION

Shannon's well known channel coding theorem (Shannon, 1948) states that for every channel a code exists, such that encoded messages can be transmitted and decoded with an error as low as needed while the transmission rate is below the channel's capacity. For practical applications, latency and computational complexity constrain code size. Thus, *structured codes* with low complexity encoding and decoding schemes, were devised.

Some structured codes possess a main feature known as the *permutation group* (PG). The permutations in PG map each codeword to some distinct codeword. This is crucial to different decoders, such as the parallelizable soft-decision Belief Propagation (BP) (Pearl, 2014) decoder. It empirically stems from evidence that whereas decoding various corrupted words may fail, decoding a permuted version of the same corrupted words may succeed (Macwilliams, 1964). For instance, this is exploited in the mRRD (Dimnik & Be'ery, 2009) and the BPL (Elkelesh et al., 2018) algorithms, which perform multiple runs over different permuted versions of the same corrupted codewords by trading off complexity for higher decoding gains.

Nonetheless, there is room for improvement since not all permutations are required for successful decoding of a given word: simply a fitting one is needed. Our work deals with obtaining the best fit permutation per word, by removing redundant runs which thus preserve computational resources. Nevertheless, it remains unclear how to obtain this type of permutation as indicated by the authors in (Elkelesh et al., 2018) who stated in their Section III.A, *"there exists no clear evidence on which graph permutation performs best for a given input"*. Explicitly, the goal is to approximate a function mapping from a single word to the most probable-to-decode permutation. While analytical derivation of this function is hard, advances in the machine learning field may be of use in the computation of this type of function.

The recent emergence of Deep Learning (DL) has demonstrated the advantages of Neural Networks (NN) in a myriad of communication and information theory applications where no analytical solutions exists (Simeone, 2018; Zappone et al., 2019). For instance in (Belghazi et al., 2018), a tight lower bound on the mutual information between two high-dimensional continuous variables was estimated with NN. Another recurring motive for the use of NN in communications has to do with the amount of data at hand. Several data-driven solutions were described in (Caciularu & Burshtein, 2018; Lin et al., 2019) for scenarios with small amounts of data, since obtaining data samples in

the real world is costly and hard to collect on-the-fly. On the other hand, one should not belittle the benefits of unlimited simulated data, see (Be'ery et al., 2020; Simeone et al., 2020).

Lately, two main classes of decoders have been put forward in machine learning for decoding. The first is the class of *model-free* decoders employing neural network architectures as in (Gruber et al., 2017; Kim et al., 2018). The second is composed of *model-based* decoders (Nachmani et al., 2016; 2018; Doan et al., 2018; Lian et al., 2019; Carpi et al., 2019) implementing parameterized versions of classical BP decoders. Currently, the model-based approach dominates, but it suffers from a regularized hypothesis space due to its *inductive bias*.

Our work leverages permutation groups and DL to enhance the decoding capabilities of constrained model-based decoders. First, a self-attention model (described in Section 3) (Vaswani et al., 2017) is employed to embed all the differentiated group permutations of a code in a word-independent manner, by extracting relevant features. This is done *once* before the test phase during a preprocess phase. At test time, a trained NN accepts a corrupted word and the embedded permutations and predicts the probability for successful decoding for each permutation. Thereafter, a set of either one, five or ten most-probable-to-decode permutations are chosen, and decoding is carried out on the permuted channel words rather than decoding an arbitrary dataset with all permutations, and empirically choosing the best subset of them. Our method is evaluated on the renowned BCH code.

## 2 RELATED WORK

Permutation decoding (PD) has attracted renewed attention (Kamenev et al., 2019; Doan et al., 2018; Hashemi et al., 2018) given its proven gains for 5G-standard approved polar codes. (Kamenev et al., 2019) suggested a novel PD method for these codes. However the main novelty lies in the proposed stopping criteria for the list decoder, whereas the permutations are chosen in a random fashion. The authors in (Doan et al., 2018) presented an algorithm to form a permutation set, computed by fixing several first layers of the underlying structure of the polar decoder, and only permuting the last layers. The original graph is included in this set as a default, with additional permutations added during the process of a limited-space search. Finally we refer to (Hashemi et al., 2018) which proposes a successive permutations scheme that finds suitable permutations as decoding progresses. Again, due to the exploding search space, they only considered the cyclic shifts of each layer. This limited-search first appeared in (Korada, 2009).

Most PD methods, like the ones mentioned above, have made valuable contributions. We, on the other hand, see the choice of permutation as the most integral part of PD, and suggest a pre-decoding module to choose the best fitting one. Note however that a direct comparisons between the PD model-based works mentioned and ours are infeasible.

Regarding model-free approaches, we refer in particular to (Bennatan et al., 2018) since it integrates permutation groups into a model-free approach. In that paper, the decoding network accepts the syndrome of the hard decisions as part of the input. This way, domain knowledge is incorporated into the model-free approach. We introduce domain knowledge by training the permutation embedding on the parity-check matrix *and* accepting the permuted syndrome. Furthermore, each word is chosen as a fitting permutation such that the sum of LLRs in the positions of the information-bits is maximized. Note that this approach only benefits model-free decoders. Here as well comparisons are infeasible.

## 3 BACKGROUND

**Coding** In a typical communication system, first, a length $k$ binary message $\boldsymbol{m} \in \{0, 1\}^k$ is encoded by a generator matrix $\boldsymbol{G}$ into a length $n$ codeword $\boldsymbol{c} = \boldsymbol{G}^\top \boldsymbol{m} \in \{0, 1\}^n$. Every codeword $\boldsymbol{c}$ satisfies $\boldsymbol{Hc} = \boldsymbol{0}$, where $\boldsymbol{H}$ is the parity-check matrix (uniquely defined by $\boldsymbol{GH}^\top = \boldsymbol{0}$). Next, the codeword $\boldsymbol{c}$ is modulated by the Binary Phase Shift Keying (BPSK) mapping ($0 \rightarrow 1, 1 \rightarrow -1$) resulting in a modulated word $\boldsymbol{x}$. After transmission through the additive white Gaussian noise (AWGN) channel, the received word is $\boldsymbol{y} = \boldsymbol{x} + \boldsymbol{z}$, where $\boldsymbol{z} \sim N(\boldsymbol{0}, \sigma_z^2 \boldsymbol{I}_n)$.

At the receiver, the received word is checked for any detectable errors. For that purpose, an estimated codeword $\hat{\boldsymbol{c}}$ is calculated using a hard decision (HD) rule: $\hat{c}_i = 1_{\{y_i < 0\}}$. If the syndrome $\boldsymbol{s} = \boldsymbol{H}\hat{\boldsymbol{c}}$ is all zeros, one outputs $\hat{\boldsymbol{c}}$ and concludes. A non-zero syndrome indicates that channel errors occurred.

Then, a decoding function $\mathrm{dec} : \boldsymbol{y} \to \{0, 1\}^n$, is utilized with output $\hat{\boldsymbol{c}}$. One standard soft-decision decoding algorithm is Belief Propagation (BP). BP is a graph-based inference algorithm that can be used to decode corrupted codewords in an iterative manner, working over a factor graph known as the *Tanner graph*. The Tanner graph is an undirected graphical model, depicting the constraints that define the code. In these graphs, BP messages that are propagated along cycles become correlated after several BP iterations, preventing convergence to the correct posterior distribution and thus reducing overall decoding performance. We refer the interested reader to (Richardson & Urbanke, 2008) for a full derivation of the BP for linear codes, and to (Dehghan & Banihashemi, 2018) for more details on the effects of cycles in codes.

Another works (Nachmani et al., 2016; 2018) assigned learnable weights $\theta$ to the BP algorithm. This formulation unfolds the BP algorithm into a NN, referred to as weighted BP (WBP). The intuition offered was that the trained weights compensate for the short cycles (these are most performance devastating) in the Tanner graph.

**Permutation Group of a code** Let $\pi$ be a permutation on $\{1, ..., n\}$. A permutation of a codeword $\boldsymbol{c} = (c_1, ..., c_n)$ exchanges the positions of the entries of $\boldsymbol{c}$:

$$\pi(\boldsymbol{c}) = (c_{\pi(1)}, c_{\pi(2)}, ..., c_{\pi(n)})^\top.$$

A permutation $\pi$ is an *automorphism* of a given code $\mathbb{C}$ if $\boldsymbol{c} \in \mathbb{C}$ implies $\pi(\boldsymbol{c}) \in \mathbb{C}$. The group of all automorphism permutations of a code $\mathbb{C}$ is denoted $Aut(\mathbb{C})$, also referred to as the PG of the code.

Only several codes have known PGs (Guenda, 2010) such as the BCH codes, given in (MacWilliams & Sloane, 1977) [pp.233] as:

$$\pi_{\alpha,\beta}(i) = \left[ 2^\alpha \cdot i + \beta \right] \pmod{n}$$

with $\alpha \in \{1, \ldots, \log_2(n + 1)\}$ and $\beta \in \{1, \ldots, n\}$. Thus a total of $n \log_2(n + 1)$ permutations compose $Aut(\mathbb{C})$.

One possible way to mitigate the detrimental effects of cycles is by using code *permutations*. We can apply BP on the permuted received word and then apply the inverse permutation on the decoded word. This can be viewed as applying BP on the original received word with different weights on the variable nodes. Since there are cycles in the Tanner graph there is no guarantee that the BP will converge to an optimal solution and each permutation enables a different decoding attempt. This strategy has proved to yield to a better convergence and overall decoding performance gains (Dimnik & Be'ery, 2009), as observed in our experiments, in Section 5.

**Graph Node Embedding** The method we propose uses a *node embedding* technique for embedding the variable nodes of the code's Tanner graph, thus taking the code structure into consideration. Specifically, in Sec. 4.2 we employ the *node2vec* (Grover & Leskovec, 2016) method. We briefly describe this method and the reader can refer to the paper for more technical details. The task of *node embedding* is to encode nodes in a graph as low-dimensional vectors that summarize their relative graph position and the structure of their local neighborhood. Each learned vector corresponds to a node in the graph, and it has been shown that in the learned vector space, geometric relations are captured; e.g., interactions that are modeled as edges between the nodes in the graph. Specifically, *node2vec* is trained by maximizing the mean probability of the occurrence of subsequent nodes in fixed length sampled random walks. It employs both breadth-first (BFS) and depth-first (DFS) graph searches to produce high quality informative node representations.

**Self-Attention** An attention mechanism for neural networks that was designed to enable neural models to focus on the most relevant parts of the input. This modern neural architecture allows for the use of weighted averaging to optimize a task objective and to deal with variable sized inputs. When feeding an input sequence into an attention model, the resulting output is an embedded representation of the input. When a single sequence is fed, the attentive mechanism is employed to attend to all positions within the same sequence. This is commonly referred to as the *self-attention* representation of a sequence. Initially, self-attention modelling was used in conjunction with recurrent neural networks (RNNs) and convolutional neural networks (CNNs) mostly for natural language processing (NLP) tasks. In (Bahdanau et al., 2015), this setup was first employed and was shown to produce superior results on multiple automatic machine translation tasks.

In this work we use self attention for permutation representation. This mechanism enables better and richer permutation modelling compared to a non-attentive representation. The rationale behind

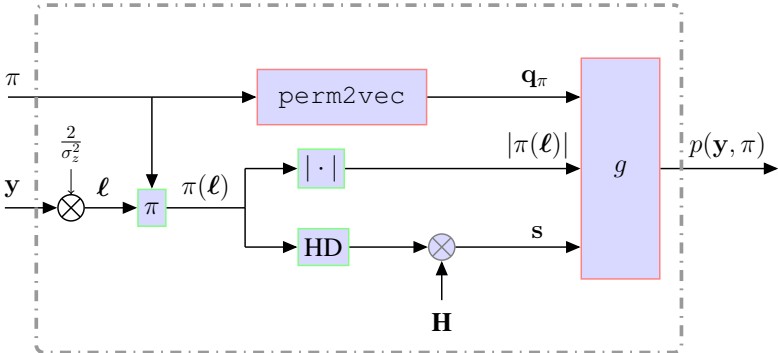

Figure 1: A schematic architecture of the Graph Permutation Selection (GPS) classifier.

using self-attention comes from permutation distance metrics preservation; a pair of "similar" permutations will have a close geometric self-attentive representation in the learned vector space, since the number of index swaps between permutations only affects the positional embedding additions.

## 4 THE DECODING ALGORITHM

### 4.1 PROBLEM FORMULATION AND ALGORITHM OVERVIEW

Assume we want to decode a received word $\boldsymbol{y}$ encoded by a code $\mathbb{C}$. Picking a permutation from the PG $Aut(\mathbb{C})$ may result in better decoding capabilities. However, executing the decoding algorithm for each permutation within the PG is a computationally prohibitive task especially if the code permutation group is large. An alternative approach involves first choosing the best permutation and only then decoding the corresponding permuted word.

Given a received word $\boldsymbol{y}$, the optimal single permutation $\pi^\star \in Aut(\mathbb{C})$ is the one that minimizes the bit error rate (BER):

$$\pi^\star = \arg\min_{\pi \in Aut(\mathbb{C})} \mathrm{BER}\Big(\pi^{-1}(\mathrm{dec}(\pi(\boldsymbol{y}))), \mathbf{c}\Big) \tag{1}$$

where $\boldsymbol{c}$ is the submitted codeword and BER is the Hamming distance between binary vectors.

The solution to Eq. (1) is intractable since the correct codeword is not known in the decoding process. We propose a data-driven approach as an approximate solution. The gist of our approach is to estimate the best permutation without applying a tedious decoding process for each code permutation and without relying on the correct codeword $\boldsymbol{c}$.

We highlight the key points of our approach below, and elaborate on each one in the rest of this section. Our architecture is depicted in Fig. 1. The main components are the *permutation embedding* (Section 4.2) and the *permutation classifier* (Section 4.3). First, the permutation embedding block perm2vec receives a permutation $\pi$, and outputs an embedding vector $\boldsymbol{q}_\pi$. Next, the vectors $\pi(\boldsymbol{y})$ and $\boldsymbol{q}_\pi$ are the input to the permutation classifier that computes an estimation $p(\boldsymbol{y}, \pi)$ of the probability of word $\pi(\boldsymbol{y})$ to be successfully decoded by dec. Next, we select the permutation whose probability of successful decoding is maximal:

$$\hat{\pi} = \arg\max_{\pi \in Aut(\mathbb{C})} p(\boldsymbol{y}, \pi) \tag{2}$$

and decoding is done on $\hat{\pi}(\boldsymbol{y})$. Finally the decoded word $\hat{\boldsymbol{c}} = \hat{\pi}^{-1}(\mathrm{dec}(\hat{\pi}(\boldsymbol{y})))$ is outputted.

### 4.2 PERMUTATION EMBEDDING

Our permutation embedding model consists of two sublayers: self-attention followed by an average pooling layer. To the best of our knowledge, our work is the first to leverage the benefits of the self-attention network in physical layer communication systems.

In (Vaswani et al., 2017), positional encodings are vectors that are originally compounded with entries based on sinusoids of varying frequency. They are added as input elements prior to the first self-attention layer, in order to add a position-dependent signal to each embedded token and help the model incorporate the order of the input tokens by injecting information about the relative or absolute position of the tokens. Inspired by this method and other recent NLP works (Devlin et al., 2019; Liu et al., 2019; Yang et al., 2019), we used learned positional embeddings which have been shown to yield better performance than the constant positional encodings, but instead of randomly initializing them, we first pre-train *node2vec* node embeddings over the corresponding code's Tanner graph. We then take the variable nodes output embeddings to serve as the initial positional embeddings. This helps our model to incorporate some graph structure and to use the code information. We denote by $d_w$ the dimension of the output embedding space (this hyperparameter is set before the node embedding training). It should be noted that any other node embedding model can be trained instead of *node2vec* which we leave for future work. Self-attention sublayers usually employ multiple attention heads, but we found that using one attention head was sufficient. Furthermore, using more self-attention layers did improve the results either.

Denote the embedding vector of $\pi(i)$ by $\mathbf{u}_i \in \mathbb{R}^{d_w}$ and the embedding of the $i$th variable node by $\mathbf{v_i} \in \mathbb{R}^{d_w}$. Note that both $\mathbf{u}_i$ and $\mathbf{v}_i$ are learned, but as stated above, $\mathbf{v}_i$ is initialized with the output of the pre-trained variable node embedding over the code's Tanner graph. Thereafter, the augmented attention head operates on an input vector sequence, $\mathbf{W} = (\mathbf{w}_1, \ldots, \mathbf{w}_n)$ of $n$ vectors where $\mathbf{w}_i \in \mathbb{R}^{d_w}, \mathbf{w}_i = \mathbf{u}_i + \mathbf{v}$.

The attention head computes a same-length vector sequence $\mathbf{P} = (\mathbf{p}_1, \ldots, \mathbf{p}_n)$, where $\mathbf{p}_i \in \mathbb{R}^{d_p}$. Each encoder's output vector is computed as a weighted sum of linearly transformed input entries, $\mathbf{p}_i = \sum_{j=1}^{n} a_{ij}(\mathbf{V}\mathbf{w}_j)$ where the attention weight coefficient is computed using the softmax function, $a_{ij} = \frac{e^{b_{ij}}}{\sum_{m=1}^{n} e^{b_{im}}}$, of the normalized relative attention between two input vectors $\mathbf{w}_i$ and $\mathbf{w}_j$, $b_{ij} = \frac{(\mathbf{Q}\mathbf{w}_i)^\top (\mathbf{K}\mathbf{w}_j)}{\sqrt{d_p}}$. Note that $\mathbf{Q}, \mathbf{K}, \mathbf{V} \in \mathbb{R}^{d_p \times d_w}$ are learned parameters matrices.

Finally, the vector representation of the permutation $\pi$ is computed by applying the average pooling operation across the sequence of output vectors, $\mathbf{q}_\pi = \frac{1}{n} \sum_{i=1}^{n} \mathbf{p}_i$, and is passed to the permutation classifier.

## 4.3 PERMUTATION CLASSIFIER

We next describe a classifier that predicts the probability of a successful decoding given received word $\boldsymbol{y}$ and a permutation $\pi$ represented by a vector $\mathbf{q}$. It is more convenient to consider the log likelihood ratio (LLR) for soft-decoding. The LLR values in the AWGN case are given by $\boldsymbol{\ell} = \frac{2}{\sigma_z^2} \cdot \boldsymbol{y}$, and knowledge of $\sigma_z$ is assumed.

The input is passed to a neural multilayer perceptron (MLP) with the absolute value of the permuted input LLRs $\pi(\boldsymbol{\ell})$ and the syndrome $\mathbf{s} \in \mathbb{R}^{n-k}$ of the permuted word $\pi(\boldsymbol{\ell})$. We first use linear mapping to obtain $\boldsymbol{\ell}' = \boldsymbol{W}_{\boldsymbol{\ell}} \cdot |\pi(\boldsymbol{\ell})|$ and $\mathbf{s}' = \boldsymbol{W}_{\mathbf{s}} \cdot \mathbf{s}$ respectively, where $\boldsymbol{W}_{\boldsymbol{\ell}} \in \mathbb{R}^{d_p \times n}$ and $\boldsymbol{W}_{\boldsymbol{s}} \in \mathbb{R}^{d_p \times (n-k)}$ are learned matrices. Then, inspired by (Wang et al., 2018), we use the following similarity function:

$$g(\mathbf{h}) = \mathbf{w}_4^\top \varphi_3(\varphi_2(\varphi_1(\mathbf{h}))) + \mathbf{b}_4 \qquad (3)$$

where,

$$\mathbf{h} = [\mathbf{q}; \boldsymbol{\ell}'; \mathbf{s}'; \mathbf{q} \circ \boldsymbol{\ell}'; \mathbf{q} \circ \mathbf{s}'; \boldsymbol{\ell}' \circ \mathbf{s}'; |\mathbf{q} - \boldsymbol{\ell}'|; |\mathbf{q} - \mathbf{s}'|; |\boldsymbol{\ell}' - \mathbf{s}'|] . \qquad (4)$$

Here $[\cdot]$ stands for concatenation and $\circ$ stands for the Hadamard product. We also define

$$\varphi_i(\mathbf{x}) = \text{LeakyReLU}(\mathbf{W}_i \mathbf{x} + \mathbf{b}_i)$$

where $\mathbf{W}_1 \in \mathbb{R}^{9d_p \times 2d_p}$, $\mathbf{W}_2 \in \mathbb{R}^{2d_p \times d_p}$, $\mathbf{W}_3 \in \mathbb{R}^{d_p \times d_p/2}$ and $\mathbf{W}_4 \in \mathbb{R}^{d_p/2}$ are the learned matrices and $\mathbf{b}_1 \in \mathbb{R}^{2d_p}$, $\mathbf{b}_2 \in \mathbb{R}^{d_p}$, $\mathbf{b}_3 \in \mathbb{R}^{d_p/2}$ and $\mathbf{b}_4 \in \mathbb{R}$ are the learned biases respectively.

Finally, the estimated probability for successful decoding of $\pi(\boldsymbol{y})$ is computed as follows,

$$p(\boldsymbol{y}, \pi) = \sigma(g(\mathbf{h}))$$

where $g(\mathbf{h})$ is the last hidden layer and $\sigma(\cdot)$ is the sigmoid function. The Graph Permutation Selection (GPS) algorithm for choosing the most suitable permutation is depicted in Fig. 1

Table 1: Values of the hyper-parameters, permutation embedding and classifier.

| SYMBOL | DEFINITION | VALUE | SYMBOL | DEFINITION | VALUE |
|--------|-----------|-------|--------|-----------|-------|
| lr | Learning rate | $10^{-3}$ | - | Optimizer | Adam |
| $d_w$ | Input embedding size | 80 | $d_p$ | Output embedding size | 80 |
| - | LeakyReLU Negative slope | 0.1 | - | SNR range [dB] | $1-7$ |
| $K$ | Mini-batch size | 5000 | - | Number of mini-batches | $10^5$ |

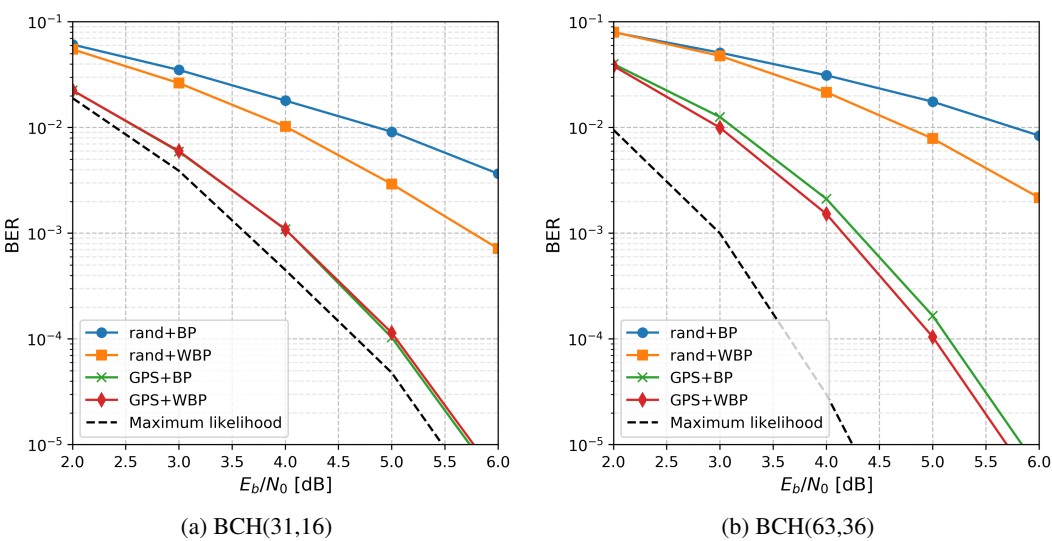

(a) BCH(31,16)

(b) BCH(63,36)

Figure 2: BER vs. SNR for GPS and random permutation selection. Both BP and WBP are considered.

### 4.4 TRAINING DETAILS

We *jointly train* the permutation embedding and the permutation classifier, employing a single decoder dec. The cross entropy loss computed for a single received word $\boldsymbol{y}$ is:

$$\mathcal{L} = -\sum_{\pi}\Big[d_{\boldsymbol{y},\pi}\log(p(\boldsymbol{y},\pi)) + (1-d_{\boldsymbol{y},\pi})\log(1-p(\boldsymbol{y},\pi))\Big]$$

where $d_{\boldsymbol{y},\pi} = 1$ if decoding of $\pi(\boldsymbol{y})$ was successful under permutation $\pi$, otherwise $d_{\boldsymbol{y},\pi} = 0$. The set of decoders dec used for the dataset generation is described in Section 5.

Each mini-batch consists of $K$ received words from the generated training dataset. This dataset contains pairs of permuted word $(\boldsymbol{y}, \pi)$ together with a corresponding label $d_{\boldsymbol{y},\pi}$. We used an all-zero transmitted codeword. Empirically, using only the all-zero word seems to be sufficient for training. Nonetheless, the test dataset is composed of randomly chosen binary codewords $\boldsymbol{c} \in \mathbb{C}$, as one would expect, without any degradation in performance. Each codeword is transmitted over the AWGN channel with $\sigma_z$ specified by a given *signal-to-noise ratio* (SNR), with an equal number of positive examples ($d$=1) and negative examples ($d$=0) in each batch. The overall hyperparameters used for training the perm2vec and the GPS classifier are depicted in Table 1.

To pre-train the node embeddings, we used the default hyperparameters suggested in the original work (Grover & Leskovec, 2016) except for the following modifications: number of random walks 2000, walk length 10, neighborhood size 10 and node embedding dimension $d_w = 80$.

Regarding computational latency, our perm2vec component is executed only at training time, which results with pretrained permutations' embeddings. Then, all the embeddings are stored in memory. At test time, we determine the probability of a permutation to decode $p(\boldsymbol{y}, \pi)$ with a single forward pass of the permutation classifier. To find the most suitable permutation, one has to compute $n \log_2(n + 1)$ such forward passes. These computations are not dependant, hence they can be done

Table 2: A comparison of the BER negative decimal logarithm for three SNR values [dB]. Higher is better. We bold the best results and underline the second best ones.

| BCH $(n,k)$ | rand+BP | | | rand+WBP | | | GPS + BP | | | GPS + WBP | | |
|---|---|---|---|---|---|---|---|---|---|---|---|---|
| SNR (dB) | 2 | 4 | 6 | 2 | 4 | 6 | 2 | 4 | 6 | 2 | 4 | 6 |
| — TOP 1 — | | | | | | | | | | | | |
| (31,16) | 1.21 | 1.74 | 2.44 | 1.26 | 1.99 | 3.14 | **1.65** | **2.96** | **5.37** | 1.65 | 2.96 | 5.31 |
| (63,36) | 1.10 | 1.51 | 2.08 | 1.10 | 1.67 | 2.66 | 1.40 | 2.67 | 5.23 | **1.42** | **2.82** | **5.44** |
| (63,45) | 1.26 | 1.90 | 2.81 | 1.25 | 2.08 | 3.67 | 1.40 | 2.58 | 5.01 | **1.42** | **2.73** | **5.35** |
| (127,64) | 0.99 | 1.30 | 1.74 | 0.99 | 1.32 | 2.11 | 1.01 | 1.94 | 4.04 | **1.01** | **1.98** | **4.14** |
| — TOP 5 — | | | | | | | | | | | | |
| (31,16) | 1.49 | 2.55 | 4.17 | 1.43 | 2.52 | 4.12 | **1.72** | **3.12** | **5.59** | 1.69 | 3.09 | 5.57 |
| (63,36) | 1.18 | 2.04 | 3.36 | 1.18 | 2.12 | 3.84 | 1.47 | 2.96 | 5.78 | **1.49** | **3.11** | **6.07** |
| (63,45) | 1.33 | 2.41 | 4.26 | 1.30 | 2.48 | 4.91 | 1.45 | 2.85 | 5.65 | **1.45** | **2.98** | **5.92** |
| (127,64) | 0.99 | 1.49 | 2.66 | 0.99 | 1.51 | 2.88 | 1.01 | 2.10 | 4.62 | **1.02** | **2.11** | **4.70** |

in parallel. To conclude, the overall computational latency of our scheme is of a single forward pass through the permutation classifier network.

## 5 EXPERIMENTAL SETUP AND RESULTS

The proposed GPS algorithm is evaluated on four different BCH codes - $(31, 16)$, $(63, 36)$, $(63, 45)$ and $(127, 64)$. As for the decoder $\mathrm{dec}$, we applied GPS on top of the BP (**GPS+BP**) and on top of a pre-trained WBP (**GPS+WBP**), trained with the same configuration taken from (Nachmani et al., 2017). All decoders are tested with 5 BP iterations and the syndrome stopping criterion is adopted after each iteration. These decoders are based on the *systematic* parity-check matrices, $\boldsymbol{H} = [\boldsymbol{P}^\top | \boldsymbol{I}_{n-k}]$, since these matrices are commonly used. For comparison, we employ a random permutation selection (from the PG) as a baseline for each decoder - **rand+BP** and **rand+WBP**. In addition, we depict the **maximum likelihood** results, which are the theoretical lower bound for each code (for more details, see (Richardson & Urbanke, 2008, Section 1.5)).

**Performance Analysis** We assess the quality of our GPS using the BER metric, for different SNR values [dB] when at least 1000 error words occurred. Note that we refer to the SNR as the normalized SNR ($E_b/N_0$), which is commonly used in digital communication. Fig. 3 presents the results for BCH(31,16) and BCH(63,36) and Table 2 lists the results for all codes and decoders, with our GPS method and random selection. For clarity, in Table 2 we present the BER negative decimal logarithm only for the baselines, considered as the *top*-1 results. As can be seen, using our preprocess method outperforms the examined baselines. For BCH(31,16) (Fig. 2a), `perm2vec` together with BP gains up to 2.75 dB as compared to the random BP and up to 1.8 dB over the random WBP. Similarly, for BCH(63,36) (Fig. 2b), our method outperforms the random BP by up to 2.75 dB and by up to 2.2 dB with respect to WBP. We also observed a small gap between our method and the maximum likelihood lower bound. The maximal gaps are 0.4 dB and 1.4 dB for BCH(31,16) and BCH(63,36), respectively.

**Top-$\kappa$ Evaluation** In order to evaluate our classifier's confidence, we also investigated the performance of the top-$\kappa$ permutations – this method could be considered as a list-decoder with a smart permutation selection. This extends Eq. (2) from top-1 to the desired top-$\kappa$. The selected codeword $\hat{\boldsymbol{c}}^\star$ is chosen from a list of $\kappa$ decoders by $\hat{\boldsymbol{c}}^\star = \arg\max_\kappa ||\boldsymbol{y} - \hat{\boldsymbol{c}}_\kappa||_2^2$, as in (Dimnik & Be'ery, 2009).
The results for $\kappa \in \{1, 5\}$ are depicted in Table 2 and Fig. 3a. Generally, as $\kappa$ increases better performance is observed, with the added-gain gradually eroded. Furthermore, we plot the empirical **BP lower bound** achieved by decoding with a 5-iterations BP over all $\kappa = n \log_2(n + 1)$ permutations; and selecting the output word by the $argmax$ criterion mentioned above. In Fig. 3a the reported results are for BCH(63,45). We observed an improvement of 0.4 dB between $\kappa = 1$ and $\kappa = 5$ and

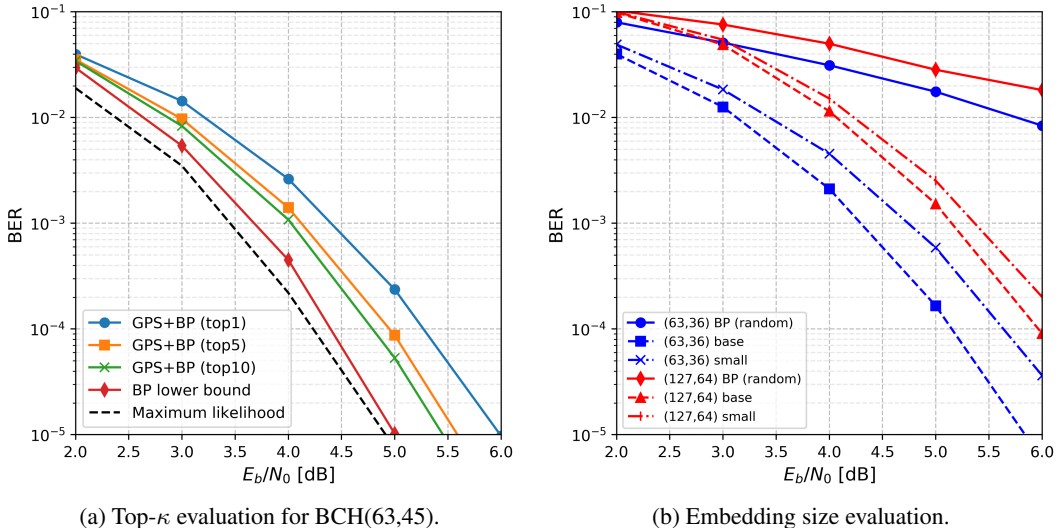

(a) Top-$\kappa$ evaluation for BCH(63,45).

(b) Embedding size evaluation.

Figure 3: BER vs. SNR performance comparison for various experiments and BCH codes.

only 0.2 dB between $\kappa = 5$ and $\kappa = 10$. Furthermore, the gap between $\kappa = 10$ and the BP lower bound is small (0.4 dB). Note that using the BP lower bound is impractical since each BP scales by $\mathcal{O}(n \log n)$ while our method only scales by $\mathcal{O}(n)$. Moreover, in our simulations, we found that the latency for five BP iterations was 10-100 times greater compared to our classifier's inference.

**Embedding Size Evaluation** In Fig. 3b we present the performance of our method using two embedding sizes. We compare our **base** model, that uses embedding size $d_q = 80$ to the **small** model that uses embedding size $d_q = 20$ (note that $d_q = d_w$). Recall that changing the embedding size also affects the number of parameters in $g$, as in Eq. (3). Using a smaller embedding size causes a slight degradation in performance, but still dramatically improves the random BP baseline. For the shorter BCH(63,36), the gap is 0.5 dB and for BCH(127,64) the gap is 0.2 dB.

**Ablation Study** We present an analysis over a number of facets of our permutation embedding and classifier for BCH (63,36), (63,45) and (127,64). We fixed the BER to $10^{-3}$ and inspected the SNR degradation of various excluded components with respect to our complete model. We present the ablation analysis for our permutation classifier and permutation embedding separately. Regarding the permutation classifier, we evaluated the complete classifier (described in Section 4.3) against its three partial versions; Omitting the permutation embedding feature vector $\mathbf{q}_\pi$ caused a performance degradation of 1.5 to 2 dB. Note that the permutation $\pi$ still affects both $\ell'$ and $\mathbf{s}'$. Excluding $\ell'$ or $\mathbf{s}'$ caused a degradation of 1-1.5 and 2.5-3 dB, respectively. In addition, we tried a simpler feature vector $\mathbf{h} = [\mathbf{q};\ell';\mathbf{s}']$ which led to a performance degradation of 1 to 1.5 dB. Regarding the permutation embedding, we compared the complete perm2vec (described in Section 4.2) against its two partial versions: omitting the self-attention mechanism decreased performance by 1.25 to 1.75 dB. Initializing the positional embedding randomly instead of using node embedding also caused a performance degradation of 1.25 to 1.75 dB. These results illustrate the advantages of our complete method, and, as observed, the importance of the permutation embedding component. Note that we preserved the total number of parameters after each exclusion for fair comparison.

## 6 CONCLUSION

We presented a self-attention mechanism to improve decoding of linear error correction codes. For every received noisy word, the proposed model selects a suitable permutation out of the code's PG without actually trying all the permutation based decodings. Our method pre-computes the permutations' representations, thus allowing for fast and accurate permutation selection at the inference phase. Furthermore, our method is independent of the code length and therefore is considered scal-

able. We demonstrate the effectiveness of `perm2vec` by showing significant BER performance improvement compared to the baseline decoding algorithms for various code lengths. Future research should extend our method to polar codes, replacing the embedded Tanner graph variable nodes by embedded factor graph variable nodes.

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
