# OpenReview forum: "Graph Permutation Selection for Decoding of Error Correction Codes using Self-Attention"
_ICLR.cc/2021/Conference — Reject_

### Official Review · AnonReviewer2 · 2020-10-28
**Methodology for optimal selection of permutations for permutation decoding**

**Rating:** 6
**Confidence:** 3

**Review:**

##### Summary
The use of permutation decoding improves the performance of some types of structured channel codes. Still, the question remains on which permutations (out of the vast set of possible ones that most of these codes allow) select to decode, in a way that we increase the chances of a successful recovery of the transmitted codeword. Obviously, we could just pass all the possible permutations through the decoder, but this is clearly extremely inefficient and hinders the usage of these methods on real-world scenarios. In the current manuscript, the authors propose a method to select the best possible permutation for every received corrupted codeword $\textbf{y}$. This allows increasing the performance in terms of error rate, and preserve the efficiency of the decoding process, up to the overhead introduced by the permutation selection procedure. The methodology is based on a permutation classification procedure (trained beforehand), where each permutation is encoded into an embedding vector obtained using the concept of self-attention, to account for the geometric similarity between transformations. The pipeline presented allows improving previous permutation decoding methods by several dBs at almost no extra cost. Hence, it renders an interesting alternative to improve further the decoding of powerful and widely used coding schemes such as Polar codes, that also accept permutation decoding.

##### Strong and weak points
Honestly, the use of self-attention for this specific task sounds quite convoluted. Still, there is no doubt it actually works, and the performance is significantly outperformed. It is unclear how this attention looks between the permutation vectors, but in the end what matters is the embedding obtained, and how it captures the similarity between them, and also, I believe, some relations with the syndrome. Still, the most important advantage is that it actually should not decrease significantly the efficiency, as the forward pass required during test could be run in parallel for all permutations, as the authors suggest. This is extremely important, and it is one of the most positive aspects of the present methodology. Besides, the paper is nicely and clearly written, with all the important details explained and all the required background information, which allows perfectly understanding the methodology.

Still, there are some important aspects, concerning the results, that may require, from my point of view, some further comparison/study (more details in the section ''Questions and additional evidence''):
* First, I do consider the comparison of GPS + (W)BP vs rand quite unfair. Is there not any other, more advanced, methods for selecting at least a subset of permutations? IN BPL, although it is applied to Polar codes, the authors finally make use of only 5 different permutations, which is not that ineffective, and enable them to report a great performance.
* The previous concern also takes me to Fig. 3a, where we can observe how there is still a lot of room for improvement, as taking the 5 top permutations still yields some extra considerable improvement.
* When performing the ablation studies, it is not clear to which of the codes those degradations correspond. I mention this because the performance degradation, for all ablations, looks quite large and pernicious, but it is difficult to assess their impact without a proper comparison. Hence, it would be interesting to see a plot showing the actual SNR values at a BER of $10^{-3}$ for some of the most harming ones, such as the exclusion of $\mathbf{s}’$. I consider depicting such comparison quite important because, in some cases, it seems that the degradation may take the model to worse performance than rand-WBP.

From my point of view, the aforementioned problems hinder the comparison of results, therefore impeding a full assessment of the scheme used, the decisions taken, and its performance compared to other similar approaches. And this is for me the weakest point of the present manuscript.

##### Decision, and key reasons
Accept, after discussing and further elaborating some of the previous concerns. Despite the previously described issues, I believe the paper presents an interesting method to advance the current state of permutation decoding. Thanks to the possibility of freely generating training samples on these schemes, it is possible to achieve very satisfactory training for permutation embedding and classification. Once that is achieved, this model can be applied at almost no extra overhead, which indeed renders the current pipeline much more useable on real-world scenarios. Additionally, the paper is nicely written and justified, easing the understanding of the methodology and the need for such an approach.


##### Questions and additional evidence
Nevertheless, I believe the authors should elaborate more on the following concerns:
* Would it be possible to propose a more fair comparison? If not, perhaps please comment on the selection of permutations done in approaches like BPL, and how that selection criterion won’t be applicable here. This may help to alleviate the feeling that comparing against rand seems a bit unfair.
* The results of Fig. 3a are quite revealing, and the room for improvement is still pretty large. Could you discuss a little bit more about the source of this error in the selection of the permutation?
* Also, and as part of some supplementary material, I would appreciate a more visual comparison of the different SNR values when performing the ablation studies. And also, discard subsets of them at the same time, etc. This will help to understand better the most key components of the pipeline, as currently it just seems everything is extremely essential, and changing any single aspect will degrade significantly the performance.
* I would like to understand why you take the absolute value of the LLR. I understand that this is the only way to proceed when you train with all $\mathbf{0}$ codewords, but is it not harming your training?
* Finally, I am curious about the Block error rate, as I can imagine that sometimes, when a permutation is quite wrongly chosen, it might push $\mathbf{y}$ to a different $\mathbf{c}$, hence leading to $\mathbf{s}=\mathbf{0}$, but still resulting in a wrong decoding. Could you provide some more insights about this?

##### Extra feedback
I just want to conclude with a suggestion for a correction and a typo:
* In Section 2, second paragraph, the sentence ''Codes with good decoding performance are represented by graphs with cycles’’ is a bit confusing, as it could be understood that indeed graphs are helping with the performance. Although you clearly explain this in the following sentence, I would rephrase that sentence.
* In page 4, when indicating the dimensions for the matrices to learn, I believe the dimensions are wrong, as they should be $\mathbf{Q}, \mathbf{K}, \mathbf{V}\in\text{R}^{d_p\times{d_w}}$.

---

> ### Author Response · Authors · 2020-11-13
> **Response to Reviewer2**
>
> Thank you for your constructive and thoughtful comments.
>
> * All of the permutations for BCH codes have (statistically) similar performance. Therefore, until now (to the best of our knowledge), there is no other work for choosing the best permutation from the permutation group, rather than random selection. Therefore, we could not provide analysis for any different better baseline.
> * First, we would like to mention that the error margin of 0.25 dB is considered small. Additionally, note that our method depends on the quality of the given decoder. Since BP has limited decoding capabilities, particularly within noisy mediums, some received words can be successfully decoded by only a small amount of permutations (sometimes even only a single one). This results in certain gaps between our method and maximum likelihood theoretical lower bound.
> * Producing graphs for ablation studies will require more time to address adequately, but we are working towards that, and this will be available in our paper if the manuscript will be accepted. Thanks for the comment.
> * As you correctly mentioned, not using the absolute values of the LLRs requires training over the entire codebook, which consists of $2^n$ possible words and is not practical at all. Despite that, we tried this technique for very short codes and it did not yield better results than the proposed training scheme. Moreover, for longer codes (e.g., $n>32$), the training did not converge at all.
> * You are correct. Choosing a wrong permutation indeed may result in the incorrect codeword, but note that it will harm both the bit and the block error rate since wrong decoding automatically results in wrong bits. Plotting also block error rate is redundant since it is highly correlated with bit error rate, and also not a common practice in other related papers. Note that we are using the widely used syndrome stopping condition to reduce the average latency (and also the complexity).
>
> Regarding you extra feedback:
>
> * Low rate codes with dense matrices (and thus a high number of short cycles) tend to have a better maximum-likelihood performance. On the other hand, the BP decoder does not perform well for codes with many short cycles and is suboptimal in these cases. Our method aims at bridging this gap and providing the BP decoder better permutations to handle this problem. We will rephrase this sentence accordingly.
> * Fixed, thanks.

---

### Official Review · AnonReviewer1 · 2020-10-28
**This paper discusses new applications of self-attention mechanisms to permutation decoding problems and shows the approach's goodness with four different BCH codes on AWGN channels.**

**Rating:** 5
**Confidence:** 4

**Review:**

[Main Strengths]

This paper's main strength is that the background section is well-drafted, the problem statement is clear, and each algorithm block's motive is adequately referenced.

=================================================================

[Main Weaknesses]

The paper's main weakness is that it seems the motivations for choosing four different BCH codes (in Section 5) are not justified clearly. It will make the authors claim stronger if they explain why these represent linear codes cases (unless the permutation decoding only applicable to the BCH codes). Additionally, the SNR gains from baselines (rand+BP) are shown large for the BCH codes (e.g., Figure 2). However, the principal question is that one will observe similarly large gains on the other capacity-achieving codes (e.g., LDPC codes), in particular, when the baseline performs better.

=================================================================

[Technical Comments]

1) Can this approach be extendable to other types of channels (e.g., channels with memory)?
2) Does the algorithm require LLR to represented as simple as the AWGN channel (in Section 3.3.)?

=================================================================

[Typographical comments]
1) For better readability, Section 4 can be moved to later (e.g., after Section 5).
2) page 2: "a self-attention model (introduced in Section 2)" should be "a self-attention model (described in Section 2)".

---

> ### Author Response · Authors · 2020-11-13
> **Response to Reviewer1**
>
> Thank you for your constructive and thoughtful comments.
>
> Answers for the technical comments:
>
> 1. Our method is independent of the channel, but one of the inputs is the log likelihood-ratio (LLR) of the received word, which either requires the channel's knowledge or a method for its estimation. AWGN channels are widely used in other neural decoder and communication systems papers and practical implementations.
> 2. The LLR is required for the permutation classification phase and not for learning the permutation representation. We further ablate this in our paper- see Section 5, removing them caused significant performance degradation.
>
> Answers for the typographical comments:
>
> 1. Fixed - the Related Work section appears now right after the introduction, as Section 2.
> 2. Fixed, thanks.

---

### Official Review · AnonReviewer4 · 2020-10-29
**Graph Permutation Selection for ECC Decoding - Review**

**Rating:** 5
**Confidence:** 4

**Review:**

The focus of the paper is error correction codes. Some structured codes have a permutation group (PG) that maps codewords via permutations. The PG structure is useful for some soft decoding algorithms (e.g., belief propagation, message passing) because of the observation that while the decoding of a codeword may fail, it often happens that the decoding of a permutation of the same codeword succeeds. The paper is concerned with obtaining the best permutation to facilitate such soft decoding, through the means of deep learning.

In order to select the most promising permutation without running the soft decoding algorithms, two main ingredients are used: (1) node embeddings of the Tanner graph of the code via the node2vec method, (2) a self attention mechanism that preserves the similarity between two permutations through the node2vec transformation, resulting in close geometric representations. The node2vec embedding is pretrained on the Tanner graph of the code. A permutation classifier is trained jointly at the same time, outputting the probability that a given codeword and permutation will decode correctly.

I find the construction proposed in the paper quite interesting. However, except for some intuitive reasons it is not clear that the approach would have practical merit. The experimental evaluation should provide a convincing argument.

I am leaning towards rejecting the paper, because I find the experimental evaluation quite limited. Unfortunately, it seems that the experiments were run on a very limited set of codes. Only BCH codes were used for evaluation. The size of the codes is really small (n = 31, 63, 127). The rate of the codes is very low, for three of them ½, for one ¾. The size of the permutation group given in the paper is n*log2(n+1) (pg. 2). Considering Table 1 we see that 100,000 mini-batches of size 5000 each have been used for training; is the network learning only because the space is relatively small and the number of training examples very large?

In the conclusion we read “our method is independent of the code length and therefore is considered scalable”. It would be great to see evidence that the method scales to large code lengths (thousands of bits). Would the learning algorithm still produce significant results? Moreover, how does the method perform on codes with high rate?

Minor comments:
- It seems that WBP is not defined (is it weighted BP?)

---

> ### Author Response · Authors · 2020-11-13
> **Response to Reviewer4**
>
> Thank you for your constructive and thoughtful comments.
>
> * As mentioned in Section 5 and will be clarified upon acceptance, since our method has relatively low complexity and can be parallelized, therefore it has much low latency, we believe our work is very beneficial for both academic and industrial purposes.
>
> * We focused on low-rate BCH codes for two reasons: (1) High-rate codes tend to have sparser parity check matrices; therefore, their BP performance is better. This makes decoding easier. (2) The respective BCH codes that we used in our evaluations are widely used in other neural decoder papers. The reason for using short length codes stems from the same reasons. Note that the weighted BP (WBP) decoder (Nachmani et al., 2017) is neural and hence does not support large codes due to memory issues, but an improved performance using our method is also expected on larger codes. Note that upon acceptance, we will add a similar short analysis for polar codes within the extra page. Also, note that our method can improve any Tanner-graph based code with a permutation group and is not limited to BCH codes. We also simulated (256,163) code for only BP (since it is too demanding for WBP in terms of space) and received promising results. We will add this upon publication.
>
> * We tried using fewer training examples, but using the reported number of examples yielded the best performance results. Note that we train using only the zero codeword. Additionally, we can produce an infinite number of training samples (use the same zero codeword with any Gaussian noise realization), so there is no reason for restricting the number of training examples.
>
> * We elaborated more on the weighted BP (WBP) decoding algorithm in the background section thanks.

---

### Official Review · AnonReviewer3 · 2020-10-30
**Nice idea, but concerns about the justification of usefulness**

**Rating:** 4
**Confidence:** 3

**Review:**

Summary:
This paper suggests a new decoding algorithm of linear error corrections codes based on self-attention.

Reasons for score:
The main idea of using self-attention for decoding linear error correction codes is interesting.
My main concern is about the justification of the usefulness of the proposed scheme.

Concerns:
- My main concern is that there is an insufficient amount of reasoning to explain why we use the proposed method over the existing decoding schemes. It would be better to compare the error performance of the proposed method over the other state-of-the-art decoding methods. For example, Fig. 3 (a) contains the BP lower bound and ML bound. I like this point, but I think the authors need to show the performance of the other existing schemes (other than the "baseline decoders" of the current manuscript) altogether to easily show the usefulness of the proposed method. Also, it would be better to clearly explain the complexity/cost of using the proposed scheme.

- In the abstract, the authors state that "the lack of theoretical insights currently impedes the exploitation of the full potential of these algorithms". For me, this sentence seems important as it might be directly followed by the statements that explain the usefulness of the proposed algorithm. However, it is hard to find for me whether the theoretical foundation of this decoding algorithm is more robust than the other decoding schemes.

- For me, it is not clear why the top k (k>1) performance matters to measure the quality of the decoding algorithm of error correction codes for communication. We usually consider the top 1 performance only for this kind of application which generally requires a very low level of error rates. Instead of plotting the top k (>1) performance, it would be better to include the comparison of performance over the best existing decoding schemes (in terms of top 1 performance given a reasonable complexity).

---

> ### Author Response · Authors · 2020-11-13
> **Response to Reviewer3**
>
> Thank you for your constructive and thoughtful comments.
>
> * Our solution could be used with any hard-decision decoder. We chose the BP decoder since it is highly popular both in academia and in the industry, and it decodes soft values, offering higher gains than a hard decision algorithm. The ordered statistic decoder (OSD), for example, suffers from high complexity (about $\mathcal{O}(n^3)$) compared to BP ($\mathcal{O}(n \log n)$, for a fixed number of iterations). This makes BP practical for industrial purposes, compared to all other decoders, since it allows low complexity and latency decoding. Note that our method serves as an extension to any existing decoder and can be applied for OSD. However, BP decoders are known for their sub-optimal baseline performance, but with relatively lower latency and complexity. For that reason, investigating BP based methods and trying to achieve near-ML performance using them is highly beneficial both for academic and industrial purposes. Please note that our top-$\kappa$ evaluation simulates list decoding (see below). We will add this note to our paper. Note that we briefly discuss the complexity at the end of Section 3.4. We plan to use the extra page to add details regarding polar codes and discuss the complexity.
> * Please note that we do not change the basic decoding scheme but use a permutation selection on top of any decoder - our method serves as an extension to any existing decoder. The sentence you mentioned, together with the fact that the performance improved dramatically, highlights the contributions and the value of our work.
> * As noted previously, the top-$\kappa$ evaluation simulates a standard existing decoding scheme, which is also known as "list decoding". We added this clarification to the manuscript, thanks.

---

### Official Review · AnonReviewer5 · 2020-11-06
**Novel application of self-attention mechanism to improve permutation decoding**

**Rating:** 6
**Confidence:** 4

**Review:**

The paper focuses on improving the computational complexity of permutation decoding. In permutation decoding, one aims to decode a permutation of the received codeword in the hope that it will lead to successful decoding as compared to applying the decoding algorithm on the received codeword. In practice, one performs decoding with multiple permutations to identify the ones which lead to successful decoding. This paper proposes a classifier that given a received codeword and permutation pair predicts whether the permutation will lead to successful decoding. This alleviates the need to perform the entire decoding with permutations to identify the successful permutations. The classifier itself relies on an embedding model for the permutations which utilizes a self-attention mechanism to embed permutations.

Overall, I think that the paper makes good contributions to the growing field of ML for communications. The utilization of self-attention to this area is novel (to the best of my knowledge). My major concern is about the comprehensiveness of the paper (see below) and the quality of the presentation. The latter should be addressable with minor revision.


Pros:

1.  The paper utilizes the self-attention mechanism to improves the computational complexity of permutation decoding. To the best of my knowledge, this is the first application of self-attention in ML for communications.

2. The paper empirically demonstrates the utility of their proposed method on BCH codes.

3. The paper performs an ablation study to showcase the importance of different components of their solutions.

Cons/Comments:

1. The paper does not present any broader ML techniques that might be useful beyond the immediate scope of the paper. If the authors feel otherwise, please include a discussion on this.

2. Even though the proposed solution can be potentially employed to decode various codes, the treatment in the paper is restricted to the BCH code. The authors should make this point clear in the abstract/introduction.  Otherwise, please include experimental results with other codes or add a discussion on this.

3. There is some room for improvement in the presentation of the paper. E.g.,

a. Make the scope of the contribution precise (see the comment above)

b. The discussion in Section 3.2 can be made more clear. How is the same node embedding v being used for all $i$ in $w_i = u_i + v$?

c. $d$ is being used to represent the embedding dimension as well as an indicator variable in the definition of the loss (Section 3.4)?

d. The computational complexity discussion at the end of Section 3.4 can be improved.

Question:

Do authors restrict themselves to a single layer of self-attention mechanism? Have the authors explored the utility of multi-layer self-attention mechanisms, as usually practiced in the ML literature?

######## Post rebuttal ##############

Thank you for your response. After going through other reviewers' comments and the authors' responses to those, I am comfortable with my original score.

---

> ### Author Response · Authors · 2020-11-13
> **Response to Reviewer5**
>
> Thank you for your constructive and thoughtful comments.
>
> As far as we are aware, there is no work done in the past that proposed a technique to represent a permutation. Our main contribution aims to provide a justified framework to map permutations to a low-dimensional space (i.e., permutation embedding) and use these representations on a downstream task. Besides, in the paper, we integrate domain knowledge into our model (e.g., syndrome) and not merely took off-the-shelf model, which dramatically improved the performance, as can be inferred from our ablation study. Additionally, our method can be utilized for any graph-based and permutation-based problems, e.g., basket completion and specific knowledge graph problems.
>
> We revised our abstract and introduction accordingly and added that we intend to evaluate BCH codes in our paper. However, note that upon acceptance, we will add a similar short analysis for polar codes within the extra page. Also, note that our method can improve any Tanner-graph based code with a permutation group and is not limited to BCH codes.
>
> a. We revised the description of our contributions accordingly, thanks.
>
> b. We revised the discussion in Section 3.2. We had there a typo: $v$ should be $v_i$ and represents the i’th variable node.
>
> c. Indeed d refers to the label, but since y is already defined in our paper, we used d with two subscripts - one for the word $y$ and one for the permutation $\pi$.
>
> d. We improved the computational complexity discussion.
>
> Regarding your last question, we do not restrict the permutation embedding to a single layer of self-attention, and indeed tried it out - this did not result in any performance improvements. We mentioned that regarding additional heads in Section 3.2, but we revised this sentence to indicate the addition of layers.

---

### Author Response · Authors · 2020-11-13
**General response for all the reviewers**

We thank all of you for your insightful comments. We updated the paper to account for some of them. A few comments will require more time to address appropriately and will be available upon publication.

We address individual reviewers' comments in the responses to their reviews.

Note that we have changed the paper's structure - the Related Work section appears now right after the introduction, as Section 2. Our rebuttal refers to the sections that appear in the previous paper (before the revision).

---

### Decision · Program_Chairs · 2021-01-07
**Final Decision**

**Decision:**

Reject

**Comment:**

The reviewers positively valued the proposed idea of performing permutation selection in permutation decoding via combining node embedding and self-attention, which seems to be of high originality. I found that this paper is mostly clearly written, except Section 3.2 as AnonReviewer5 commented. The main concern among the reviewers is regarding applicability of the proposal beyond the BCH codes.

Pros:
- The proposal of utilizing self-attention for permutation selection in permutation decoding is novel and interesting.
- Computational complexity in the decoding phase is only slightly increased compared with random permutation selection, and is far smaller than performing decoding for all permutations. The GPS classifier can be parallelizable to further reduce latency.
- The proposal should be applicable beyond the BCH codes to those with decoding based on the Tanner graph, including polar codes.

Cons:
- Only the BCH codes were considered, whereas in the authors responses they will add a short analysis on polar codes.
- It seems that systematic enumeration of the PG is required, which would limit applicability of the proposal.
- There is a room for improvement in presentation:
-- In Section 3.2, the description of "positional encodings" was unclear to me, in that the ordering of the codeword entries is arbitrary, unlike typical sequence transduction problems to which attention mechanism is being applied.
-- Dependence of the input vector sequences of the attention head on the permutation $\pi$ is not clearly explained.
-- Performance of the proposed method might depends on choices of the parity-check matrix, which is however not discussed in this paper at all.

Based on the above concerns, the paper is not yet ready for publication in its current form.

Minor points:
In page 3, line 14, "that" should be deleted.
In references list, "hdpc" should be in capital. "reed-muller" should be "Reed-Muller".